# The Role of Parental Adherence to the Mediterranean Diet and Family Time Together in Children’s Weight Status: The BeE-School Project

**DOI:** 10.3390/nu16070916

**Published:** 2024-03-22

**Authors:** Ana Duarte, Juliana Martins, Maria José Silva, Cláudia Augusto, Silvana Peixoto Martins, Rafaela Rosário

**Affiliations:** 1Health Sciences Research Unit: Nursing (UICISA: E), Nursing School of Coimbra, Avenida Bissaya Barreto, Polo C, 3046-851 Coimbra, Portugal; julianacmmartins18@gmail.com (J.M.); mjsilva@ese.uminho.pt (M.J.S.); coliveira@ese.uminho.pt (C.A.); silvana.martins12@gmail.com (S.P.M.); rrosario@ese.uminho.pt (R.R.); 2School of Nursing, University of Minho, Edifício 4, Campus de Gualtar, 4710-057 Braga, Portugal; 3Research Centre on Nursing (CiEnf), School of Nursing, University of Minho, Edifício 4, Campus de Gualtar, 4710-057 Braga, Portugal; 4Research Centre on Child Studies (CIEC-UM), University of Minho, Campus de Gualtar, 4710-057 Braga, Portugal; 5ProChild CoLAB Against Poverty and Social Exclusion Association, Campus de Couros, Rua Vila Flor, 166, 4810-225 Guimarães, Portugal

**Keywords:** MEDAS, childhood, family, health promotion

## Abstract

The family context has been associated with children’s weight status. This study aims to investigate the association of parents’ adherence to the Mediterranean diet and family time with the weight status of children. The research is part of BeE-school, a cluster-randomized trial implemented in primary schools located in socially vulnerable contexts. A total of 735 children (380 boys and 355 girls) aged 6 to 10 participated in the study. Anthropometrics were assessed during school time, and weight status was categorized, while parents self-reported sociodemographic variables, adherence to the Mediterranean diet (MEDAS questionnaire), and family time. Children from families with higher education levels whose parents have a high adherence to the Mediterranean diet have lower odds of overweight/obesity (odds ratio (OR) 0.301, 95% CI 0.143–0.634, *p* = 0.002). Also, children from families with lower education levels who have more time together with their family have lower odds of overweight/obesity (OR 0.731, 95% CI 0.573–0.934, *p* = 0.012). The family environment, mainly family time together and adherence to the Mediterranean diet, exerts a significant influence on children’s weight status. Professionals working in children’s health should consider the family when fostering health-promoting behaviors.

## 1. Introduction

The positive impact of adopting a Mediterranean diet has been emphasized in recent years [1]. This diet is widely recognized as one of the healthiest dietary patterns, offering substantial health benefits, including improvements in both classical and emerging cardiovascular risk factors within just three months of its adoption [2,3] Additionally, it has been extensively associated with an overall higher quality of life [4].

The benefits of the Mediterranean diet are linked to various factors, with one key element being the consumption of extra virgin olive oil as the primary added fat [5]. In their narrative review, Flynn and colleagues [5] identified that the consumption of extra virgin olive oil provides superior benefits in managing clinical biomarkers, including weight management and glycemic control. These protective effects of extra virgin olive oil appear to be associated with its polyphenol content rather than its monounsaturated fat content, contrary to previous considerations [5]. The inclusion of mixed nuts in the Mediterranean diet also provides potential health benefits [6]. The consumption of almonds, hazelnuts, pistachios, walnuts, and other nuts has been emphasized as a great source of magnesium and potassium [7]. Some nuts also serve as sources of zinc and selenium [7]. Specifically, almonds appear to be a source of calcium [7]. Nut consumption seems to make significant contributions to health outcomes, even during infancy. For instance, Qin and colleagues [8] reported that consuming nuts leads to beneficial changes in lipid metabolism, potentially lowering the risk of hypertension in children.

This dietary pattern typically includes daily consumption of olive oil, fruits, vegetables, nuts, legumes, and whole grains, along with weekly consumption of fish and poultry. It is characterized by relatively low consumption of red meat and moderate alcohol intake, often consumed with meals [3,9]. The consumption of these foods, especially fruits, vegetables, nuts, legumes, and whole grains, has been associated with improved health outcomes [10,11,12] and is highly recommended by major scientific associations, such as the World Health Organization [4].

The Mediterranean Diet Adherence Screener (MEDAS) was designed to evaluate adherence to the Mediterranean diet [3]. MEDAS consists of a 14-item questionnaire that assesses both the type of foods consumed and the frequency of their consumption, encompassing foods that are characteristic of the Mediterranean diet, as well as those that are not [3].

The impact of family dietary patterns on children’s eating habits and intake has been firmly documented [13]. Consequently, the crucial role of families in enhancing their children’s nutritional health by increasing the consumption of wholesome foods and decreasing the intake of low-nutrition and energy-dense items such as sweetened beverages is well established [13,14].

Not only do family dietary choices have a significant impact on children’s health, but the relationship between spending quality time together as a family and children’s health seems crucial. For instance, Waters et al. [15] explored the association between parental quality time and children’s flourishing, concluding that children whose parents have insufficient quality time experience, have lower levels of flourishing. Also, families with lower socioeconomic status and lower education levels exhibit poorer levels of flourishing in children [15,16].

Family time is acknowledged as a factor that positively influences parents’ quality of life [17] and contributes to children’s well-being [14,15]. Family time can be either active or inactive. Active family time encompasses activities such as taking a walk, visiting places, and engaging in sports, while non-active family time involves activities like sitting and talking, watching TV, playing indoor games, sharing a meal, or visiting friends [18]. The type of activity engaged in by parents appears to impact the activities undertaken by children. For instance, increased parental TV watching was linked to a higher likelihood of children engaging in elevated levels of TV viewing [19].

Children who have overweight or obesity face a heightened risk of developing and retaining excess weight into adulthood [20,21]. Additionally, obesity serves as a risk factor for cardiometabolic diseases and stands as a robust predictor of mortality [22,23]. Gaining insight into how family habits impact children’s behaviors and weight status could assist in mitigating the risk of childhood overweight [24].

Despite the well-established associations between family time, adherence to the Mediterranean diet, and children’s health outcomes [13,15], there is limited evidence about their associations in children from schools prone to vulnerability. It is possible that children attending these disadvantaged schools are at a higher risk of overweight or obesity than children from non-disadvantaged schools [25]. Thus, the aim of this study is to analyze the associations of parents’ adherence to the Mediterranean diet and family time with the weight status of children from disadvantaged schools.

## 2. Materials and Methods

### 2.1. Study Design and Ethics

This study is part of BeE-school, a cluster-randomized trial, which was implemented in primary schools prone to vulnerability. A total of 735 children (380 boys (51.7%) and 355 girls (48.3%)) between the ages of 6 and 10 were enrolled in the study and granted permission to participate. Inclusion criteria included children from ten different primary schools located in economically and socially disadvantaged territories marked by poverty and social exclusion. Children with impairments, whether cognitive or physical, that could compromise data collection were excluded from the study.

This study was approved by the Ethics Subcommittee for Life and Health Sciences (CE.CVS 009/2022) of the University of Minho and registered in the Clinical Trials database/platform (NCT05395364).

The work described herein was carried out following The Code of Ethics of the World Medical Association (Declaration of Helsinki). All participants (parents or caregivers) signed the informed consent, and the children were asked and assent to participate before the procedures.

### 2.2. Adherence to Mediterranean Diet

Adherence of parents to the Mediterranean diet was evaluated using the self-reported Mediterranean Diet Adherence Screener (MEDAS), a fourteen-item multiple-choice questionnaire designed to assess the typical consumption of Mediterranean diet components. This questionnaire was validated for the Portuguese population [3]. In the present study, parents (mother, father, or both) self-reported the MEDAS questionnaire. The portion of each food group was provided, and the research team was available to clarify any doubts by phone. Those who completed the questionnaire were designated as the ‘respondents’.

Favorable responses indicating adherence to the Mediterranean diet were assigned a score of one, while unfavorable responses were scored as zero. The final score ranged from zero to fourteen, with a MEDAS score exceeding nine indicating a high level of adherence to the Mediterranean diet [3]. We categorized Mediterranean diet adherence into two groups as follows: low adherence for scores below nine and high adherence for scores exceeding ten. This categorized variable was used in the statistical analysis.

### 2.3. Anthropometrics

Children’s weight and height were measured while at school by trained researchers using standardized procedures. Weight and height measurements were recorded using a pediatric scale/stadiometer (SECA 799), and weight was rounded to the nearest 100 g. These measurements were taken with the children lightly dressed, and they were not wearing shoes. In case of doubt or uncertainty about the measurement, the procedure was repeated.

We calculated the Body Mass Index (BMI) as the ratio of body weight to height squared (kg/m^2^), then converted it into standard deviation (SD) scores (BMI z-score) adjusted for specific age and sex categories using the WHO growth reference [26]. Weight status was defined according to WHO cutoff points and subsequently categorized into two groups as follows: normal weight (SD ≤ 1) and overweight/obesity (SD > 1).

Parents self-reported their height and weight, from which BMI was calculated. In the statistical analyses, the respondent’s BMI was used as a covariate.

### 2.4. Sociodemographic Variables

Education levels of both parents were collected through a straightforward sociodemographic questionnaire developed by the researchers. The question presented multiple-choice answers, which were subsequently categorized into two groups as follows: less than higher education and higher education level.

### 2.5. Family Time

A selected question from the Health Behaviour in School-Aged Children (HBSC) survey, as outlined by Currie et al. [16], was employed to evaluate family time. The list of shared activities among families comprises the following eight items: (1) watching TV or a video together, (2) playing indoor games together, (3) eating a meal together, (4) going for a walk together, (5) going places together, (6) visiting friends or relatives together, (7) playing sports together, and (8) sitting and talking about things together.

Parents indicated the frequency with which they engaged in these activities and spent time with their children in shared endeavors. Response options ranged from every day (coded as 5) to most days of the week (coded as 4), once a week (coded as 3), less than once a week (coded as 2), and never (coded as 1).

A higher total score indicates a greater likelihood of engaging in these activities as a family [18]. In addition, further subscales were computed as “total family time” (all eight items; scored 8 to 40), “active family time” (including items 4, 5, and 7; scored 3 to 15), and “non-active family time” (including items 1, 2, 3, 6, and 8; scored 5 to 25). These variables were standardized as z-scores for subsequent statistical analyses.

The internal consistency for the total family time scale in this study was α = 0.73, that for the active family time scale was α = 0.72, and that for the non-active family time scale was α = 0.58.

### 2.6. Statistical Analysis

Descriptive analysis included central tendency measures and dispersion according to the type of variable. For a descriptive analysis of the quantitative variables, the mean and standard deviation (SD) were calculated; for categorical variables, count (*n*) and percentages (%) were used. We also compared the proportions of categorical variables using chi-squared tests for contingency tables and Student’s *t*-test for quantitative variables.

Binary logistic regression was used to estimate the associations between parents’ adherence to the Mediterranean diet (predictor) and children’s weight status (dependent variable) and between family time (predictor) and children’s weight status (dependent variable). This analysis was conducted separately based on the level of parental education because of the significant interaction observed with adherence to the Mediterranean diet. (please see Appendix A).

Children’s sex and age, as well as the respondent’s BMI, were taken into account as covariates. The level of significance was established as <0.05. The data analysis was performed using IBM SPSS^®^, version 29.0.

## 3. Results

### 3.1. Characterization of Participants

The characteristics of the study participants are presented in Table 1. A total of 735 children were enrolled in the study (380 boys), with a mean (SD) age of 7.7 (1.2) years. Most of the children had a normal weight (61.5%), with slight differences between boys and girls (62.2% and 60.8%, respectively). There were 280 children (38.5%) with overweight/obesity. Concerning parents’ BMI, there were no significant differences between having a son or a daughter, with means (SD) of 25.3 (4.7) for mothers’ BMI and 26.5 (3.7) for fathers’ BMI.

Most mothers had a higher education level (44.2%), with no significant differences between children. In contrast, fathers’ education differed significantly between those with a participating son and those with a participating daughter (*p* = 0.008). Most fathers had a level of less than higher education (67.8%).

Most parents presented low adherence to the Mediterranean diet (79.7%). Concerning total family time, a mean (SD) score of 26.9 (4.0) was obtained. Active family time had a mean (SD) score of 7.7 (2.3), and non-active family time, scored from 5 to 25, had a mean (SD) score of 19.2 (2.4).

### 3.2. Logistic Binary Regression Results

Table 2 displays the associations between children’s weight status and their parents’ adherence to the Mediterranean diet, as well as family time, according to the highest level of parents’ education level. We conducted the analysis using three adjustment models to explore the relationship between children’s weight status and parents’ adherence to the Mediterranean diet, as well as the association of children’s weight status with total family time, active family time, and non-active family time.

A higher level of parental adherence to the Mediterranean diet was significantly associated with reduced odds of overweight children in those families with a higher education level (odds ratio (OR) 0.301, 95% CI 0.143–0.634, *p* = 0.002). Even when adjusted for potential confounders (children’s sex and age and respondent BMI), the model remained significant (OR 0.358, 95% CI 0.165–0.775, *p* = 0.009).

When we observed the association between family time and children’s weight status, we found that family time was significantly associated with children’s weight status in families with lower education levels (OR 0.731, 95% CI 0.573–0.934, *p* = 0.012).

## 4. Discussion

This study found that children from families with higher education levels whose parents have a high level of adherence to the Mediterranean diet have lower odds of overweight/obesity. Even when adjusted for potential confounders, a higher adherence to this diet appears to be a protective factor against children’s overweight in families with higher education levels. Also, we observed that children from families with lower education levels who spend more time together with their family have lower odds of overweight/obesity.

These results are in accordance with existing evidence that confirms the benefits of the Mediterranean diet for health in general and for the prevention of overweight [9,27,28]. Moreover, our findings highlight the significant association of family habits with children’s health. According to Griban and colleagues [29], the motivation behind children adopting a healthy lifestyle lies with the family.

In the present study, we did not explore the potential association between the specific consumption of olive oil, nuts, or other foods but rather focused on adherence to the Mediterranean diet. Numerous studies have confirmed the significant impact of a comprehensive pattern instead of an individual food, such as the Mediterranean diet, on health. Regardless of the primary types of food contributing to better health outcomes, this diet appears to confer overall health benefits [1,9,27,28]. An umbrella review conducted by Dinu and colleagues [28] revealed that increased adherence to the Mediterranean diet was linked to reduced risks of overall mortality, cardiovascular diseases, coronary heart disease, myocardial infarction, overall cancer incidence, neurodegenerative diseases, and diabetes. In another study conducted by Bakırhan and colleagues [27] involving a sample of 1991 children and adolescents with various physical or intellectual disabilities, it was concluded that greater adherence to the Mediterranean diet was associated with an improved quality of life.

As already mentioned, the family’s diet significantly influences the food consumption of children [13]. Therefore, parents adhering to the Mediterranean diet could influence the health outcomes of their children. In our study, higher adherence to the Mediterranean diet by parents was significantly associated with lower odds of overweight or obesity in children, even when adjusting for various confounders. Based on the literature, we chose to include children’s sex and age, parents’ highest level of education, and respondent BMI in the full model as potential variables that could influence children’s BMI. Children whose parents have overweight appear to be more likely to also have overweight [30], emphasizing the inclusion of respondent BMI in the analysis. The associations of body mass index between parents and children are explained by genetic and behavioral determinants [30,31]. Obesity is a multifactorial disease influenced by internal factors such as genetics, sex, age, and psychological characteristics. Additionally, it is influenced by external factors, including environmental, socio-cultural, and family determinants (parenting style and parents’ lifestyles), which significantly impact lifestyle, dietary choices, and physical activity [31].

We included the higher education level of both parents in the analysis, since previous research has shown that parents’ consistency in applying rules regarding dietary and sedentary behaviors is associated with their level of education [32]. Additionally, there seems to be an association between parents’ educational level and child overweight/obesity, with the lowest educational level corresponding to the highest prevalence of obese children [30]. Furthermore, in relation to socioeconomic status and diet, Gautam and colleagues [33] found that higher socioeconomic status was associated with higher consumption of breakfast, fruit, vegetables, dairy products, and a balanced diet in general. This aligns with our findings, confirming that the adoption of a Mediterranean diet among families with higher education levels promotes a healthy weight status for their children.

Concerning parents’ weight status, we found that almost half (46.9%) of respondents had overweight/obesity. These findings align with national data for Portugal, where 36.5% of adults aged 18 to 64 have overweight and 21.6% have obesity [34]. We noted a significantly low adherence to the Mediterranean diet among parents, as almost 80% reported non-adherence. This trend may be linked to elevated parental BMI, as the Mediterranean diet has been shown to be effective in reducing BMI [28]. The low adherence to the Mediterranean diet could be associated with various factors, one of which is low socioeconomic status [35]. Moustakim and colleagues [36] found that individuals from households with medium and high socioeconomic status exhibit higher adherence to the Mediterranean diet compared to those from disadvantaged households with low socioeconomic status. Additionally, a high adherence to the Mediterranean diet was linked to increased dietary expenses [37]. Given the inherent social vulnerabilities in our sample, this could serve as a plausible explanation for the low adherence to the Mediterranean diet.

Regarding family time, parents’ education level appears to be significantly associated with children’s weight status, especially for children from families with lower levels of education. The importance of shared moments in family is irrefutable. However, in recent years, family time has been increasingly reduced due to busy schedules and employment commitments of parents [38]. One of the privileged moments for families is mealtime [18]. According to Brannen and colleagues [38], this is one of the best moments for family ‘togetherness’. Additionally, mealtime serves as a moment when family members gather to share food, engage in emotional communication, and impart valuable knowledge to children [13]. This family time is classified as ‘non-active family time’, similar to activities like watching TV or a video together, playing indoor games together, visiting friends or relatives together, and sitting and talking about things together. According to our study, non-active time seems to be significantly associated with the likelihood of children developing overweight or obesity among families with lower education levels, but this was just observed in the unadjusted model.

Family time plays an important role in children’s development. For example, a study conducted by Korcz and colleagues [18] investigated the association between family time and the physical activity of children aged 10–11. They found that playing sports together and engaging in indoor games are less frequent activities that families typically engage in together, highlighting that time spent in a family setting does not always involve physical activity. Furthermore, their findings underscore the importance of family time in terms of role-modeling physical activity, providing a safe and interesting backyard for children to play in, establishing rules about screen entertainment use at home, and promoting a healthy lifestyle [18].

Our findings emphasize the importance of adopting a Mediterranean diet among families with higher education levels to promote a healthy weight status for their children. Additionally, particularly within families with lower education levels, the promotion of family time appears to be protective for children’s weight maintenance. Future health promotion programs should consider implementing strategies that address both dietary interventions, with a focus on promoting a Mediterranean diet among families with higher education levels, and the encouragement of family time, especially within families with lower education levels, as these measures appear to contribute to protective factors for children’s weight maintenance.

This study has several strengths that should be mentioned. Firstly, this study is part of a broader research project, and data collection was carried out by trained researchers, thereby enhancing the reliability of our findings. Additionally, we accounted for significant potential confounders in our data analyses, such as children’s age and sex and respondent BMI, to minimize the likelihood of spurious associations. Furthermore, this study contributes to the examination of the associations of family dietary patterns and jointly performed activities with the weight status of children.

This study also has some limitations. Firstly, the study employed a cross-sectional design with a limited sample size, thereby restricting the generalizability of the findings to other populations. Secondly, since the data on parents’ adherence to the Mediterranean diet and family time together were self-reported, their accuracy should be approached with caution. It is crucial to acknowledge that the self-report method employed in this study may introduce social desirability bias, where parents may have reported their adherence to the Mediterranean diet and family time in a more positive or socially desirable manner than what is occurring. To address potential misinformation in the MEDAS questionnaire, a caption for each item was included to provide a more detailed explanation of the meaning of possible answers. Notably, the family time questionnaire has not yet been validated for the Portuguese context.

The influence of the home setting on children’s health is important, since parents determine food availability at home and influence children’s nutrition [14]. Therefore, professionals who work in children’s health should also consider the family’s influence in supporting the development of health-promoting behaviors [39]. In conclusion, school-based interventions should underscore the crucial role of family habits in influencing children’s health, considering their socioeconomic status.

## 5. Conclusions

This study revealed that a higher education level and a higher level of adherence to the Mediterranean diet among parents were significantly associated with lower odds of their children experiencing overweight or obesity. Additionally, family time together showed a significant association with children’s weight status among families with lower education levels. The family environment plays a pivotal role in shaping children’s weight status. For instance, promoting adherence to the Mediterranean diet among families with school-aged children can contribute to children’s healthier weight status. Additionally, emphasizing quality family time together is also associated with children’s healthier weight status. Future studies should incorporate longitudinal designs to determine the sustainability of these results over time. Moreover, intervention programs ought to incorporate the family’s influence in fostering behaviors that contribute to overall health.

## Figures and Tables

**Table 1 nutrients-16-00916-t001:** Descriptive analyses.

	All (*n* = 735)	Boys (*n* = 380)	Girls (*n* = 355)	*p **
Age (mean ± SD)	7.7 ±1.2	7.7 ±1.2	7.7 ±1.2	0.559
Weight status *n* (%)				0.690
Normal weight	448 (61.5%)	234 (62.2%)	214 (60.8%)	
Overweight/obesity	280 (38.5%)	142 (37.8%)	138 (39.2%)	
Mother’s education *n* (%)				0.197
Less than higher education	339 (55.8%)	163 (53.3%)	176 (58.5%)	
Higher education	268 (44.2%)	143 (46.7%)	125 (41.5%)	
Father’s education *n* (%)				**0.008**
Less than higher education	391 (67.8%)	179 (62.6%)	212 (72.9%)	
Higher education	186 (32.2%)	107 (37.4%)	79 (27.1%)	
Parents’ highest education *n* (%)				0.168
Less than higher education	306 (50.2%)	146 (47.4%)	160 (53.0%)	
Higher education	304 (49.8%)	162 (52.6%)	142 (47.0%)	
Mother’s BMI (mean ± SD)	25.3 ±4.7	25.0 ±4.1	25.5 ±4.6	0.215
Father’s BMI (mean ± SD)	26.5 ±3.7	26.7 ±3.7	26.5 ±3.7	0.516
Respondents’ BMI (mean ± SD)	25.5 ±4.4	25.1 ±4.0	25.8 ±4.8	0.080
Respondent weight status *n* (%)				0.343
Normal	290 (53.1%)	150 (55.1%)	140 (51.1%)	
Overweight/obesity	256 (46.9%)	122 (44.9%)	134 (48.9%)	
MEDAS *n* (%)				0.065
Low adherence	388 (79.7%)	201 (83.1%)	187 (76.3%)	
High adherence	99 (20.3%)	41 (16.9%)	58 (23.7%)	
Family time raw score (mean ± SD)				
Total	26.9 ± 4.0	27.1 ± 4.0	26.7 ± 4.0	0.249
Active time	7.7 ± 2.3	7.8 ± 2.3	7.6 ± 2.2	0.233
Non-active time	19.2 ± 2.4	19.3 ± 2.3	19.1 ± 2.5	0.361

* Student’s *t*-tests (continuous variables) or chi-squared tests (categorical variables).

**Table 2 nutrients-16-00916-t002:** Binary logistic regression analysis of the associations between children’s weight status, parent’s adherence to the Mediterranean diet, and family time z-score according to parents’ education level.

		Children’s Weight Status			
		Variables	Model 1	Model 2	Model 3
**Parents’ education level**	Less than higher education	**MEDAS**			
		Low adherence	Ref.	Ref.	Ref.
		High adherence	0.916 (0.470; 1.786)	0.942 (0.476;1.864)	0.992 (0.496; 1.982)
		**Total Family Time** (z-score)	**0.731 (0.573; 0.934)**	**0.773 (0.603; 0.990)**	0.796 (0.621; 1.021)
		**Active** Family Time (z-score)	0.804 (0.627; 1.032)	0.831 (0.644; 1.073)	0.859 (0.665; 1.110)
		**Non-active** Family Time (z-score)	**0.753 (0.597; 0.950)**	0.797 (0.629; 1.011)	0.812 (0.638; 1.033)
	Higher education	**MEDAS**			
		Low adherence	Ref.	Ref.	Ref.
		High adherence	**0.301 (0.143; 0.634)**	**0.317 (0.149; 0.674)**	**0.358 (0.165; 0.775)**
		**Total Family Time** (z-score)	0.841 (0.626; 1.129)	0.856 (0.636; 1.152)	0.898 (0.660; 1.222)
		**Active** Family Time (z-score)	0.807 (0.615; 1.057)	0.803 (0.611; 1.054)	0.811 (0.611; 1.076)
		**Non-active** Family Time (z-score)	0.988 (0.737; 1.323)	1.191 (0.937; 1.513)	1.071 (0.787; 1.458)

**Model 1**—unadjusted model; **Model 2**—model adjusted for children’s sex and age; **Model 3**—model adjusted for children’s sex and age and respondent BMI. **Note:** values expressed as OR (CI 95%). ***p*-value < 0.05.**

## Data Availability

The original contributions presented in the study are included in the article/Appendix A, further inquiries can be directed to the corresponding author.

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
