# Peer review of "The Role of Parental Adherence to the Mediterranean Diet and Family Time Together in Children’s Weight Status: The BeE-School Project"

_nutrients, 2024, doi:10.3390/nu16070916_

Round 1
Reviewer 1 Report
Comments and Suggestions for Authors
Dear Authors,
The results are of interest to researchers studying children's eating behavior and health in a broad sense. However, I found some elements that require clarification or correction - Please respond to them.
line 55 "sugary" - "Sweetened" - may be better?
line 96 - MEDAS - Please provide references from the literature and the basis for validating the questionnaire.
line 109 - "Weight and height measurements were recorded..." - How many repetitions were the measurements performed?
lines 230-257 - In my opinion, the benefits of the Mediterranean Diet should not be described here. This is material for the Introduction section. Here you should focus on discussing the connections contained in the topic. The authors should focus more on the relationship between the healthiness of the diet and body weight and BMI.
line 265 - "Children whose parents have overweight appear to be more likely to also have overweight" - How can this be explained?
References - Please prepare References in accordance with the journal's requirements.

Reviewer 2 Report
Comments and Suggestions for Authors
The following observations are for the purpose of improving/completing the content of the manuscript.
ABSTRACT
- The sample size should appear for each sex: 380 boys and 355 girls.
KEYWORDS
- It is suggested to change the keywords to other terms that are not already in the title of the manuscript. In this case, "mediterranean diet" and "children".
- Lines 47-48: if MEDAS is a validated questionnaire, please indicate it clearly.
MATERIALS AND METHODS
- Lines 88-89: sample number and percentage should be shown for both boys and girls.
- Line 91: the abbreviation TEIP is not previously defined.
- Neither the inclusion criteria nor the exclusion criteria are explicit.
- Lines 135-137: for response options, an even number of possible responses (e.g., 6) would have been more reliable. In the case of an odd number, such as 5, there may be a tendency to mark the middle neutral value.
- The information regarding the approval of the corresponding Ethics Committee and the signing of informed consent by the parents should appear at the beginning of Materials and Methods, between sections 2.1 and 2.2, since it is information that must be known from the beginning. The last section should be Statistical Analysis.
- Line 160: change to "the level of significance was established as <0.05".
RESULTS
- In Table 1 the use of parentheses is confusing: in some variables it is SD, while in others it is %. This leads to confusion. Please, when it comes to SD, then the results would have to be expressed as "mean ± SD". If it is a percentage, then "%" should appear next to the numerical value. The clarification at the bottom of the table with the superscript 1 does not work as such.
- In Table 1, the significant difference in the item "Fathers education" is not marked.
DISCUSSION AND CONCLUSIONS
- Very well thought out discussion, with sufficient support of references to discuss and with a reflective character.
- Conclusions could be expanded considering the different family situations, not only those that have presented differences, to have a more global perspective of the study (of course, focusing on the most notable results).
